# Exploring the Interplay between Bone Marrow Stem Cells and Obesity

**DOI:** 10.3390/ijms25052715

**Published:** 2024-02-27

**Authors:** Fiorenzo Moscatelli, Antonietta Monda, Giovanni Messina, Elisabetta Picciocchi, Marcellino Monda, Marilena Di Padova, Vincenzo Monda, Antonio Mezzogiorno, Anna Dipace, Pierpaolo Limone, Antonietta Messina, Rita Polito

**Affiliations:** 1Department of Wellbeing, Nutrition and Sport, Pegaso Telematic University, 80143 Naples, Italy; fiorenzo.moscatelli@unipegaso.it (F.M.); anna.dipace@unipegaso.it (A.D.); pierpaolo.limone@unipegaso.it (P.L.); 2Department of Experimental Medicine, Section of Human Physiology and Unit of Dietetics and Sports Medicine, University of Campania “Luigi Vanvitelli”, 80138 Naples, Italy; antonietta.monda@unicampania.it (A.M.); giovanni.messina@unicampania.it (G.M.); elisabetta.picciocchi@unicampania.it (E.P.); marcellino.monda@unicampania.it (M.M.); 3Department of Humanistic Studies, University of Foggia, 71100 Foggia, Italy; marilena.dipadova@unifg.it; 4Department of Exercise Sciences and Well-Being, University of Naples “Parthenope”, 80138 Naples, Italy; vincenzo.monda@uniparthenope.it; 5Department of Mental Health, Fisics and Preventive Medicine, University of Campania “Luigi Vanvitelli”, 80138 Naples, Italy; antonio.mezzogiorno@unicampania.it; 6Department of Clinical and Experimental Medicine, University of Foggia, 71122 Foggia, Italy; rita.polito@unifg.it

**Keywords:** bone marrow, stem cells, obesity, mesenchymal stem cells, adipose tissue, inflammation

## Abstract

Obesity, a complex disorder with rising global prevalence, is a chronic, inflammatory, and multifactorial disease and it is characterized by excessive adipose tissue accumulation and associated comorbidities. Adipose tissue (AT) is an extremely diverse organ. The composition, structure, and functionality of AT are significantly influenced by characteristics specific to everyone, in addition to the variability connected to various tissue types and its location-related heterogeneity. Recent investigation has shed light on the intricate relationship between bone marrow stem cells and obesity, revealing potential mechanisms that contribute to the development and consequences of this condition. Mesenchymal stem cells within the bone marrow, known for their multipotent differentiation capabilities, play a pivotal role in adipogenesis, the process of fat cell formation. In the context of obesity, alterations in the bone marrow microenvironment may influence the differentiation of mesenchymal stem cells towards adipocytes, impacting overall fat storage and metabolic balance. Moreover, bone marrow’s role as a crucial component of the immune system adds another layer of complexity to the obesity–bone marrow interplay. This narrative review summarizes the current research findings on the connection between bone marrow stem cells and obesity, highlighting the multifaceted roles of bone marrow in adipogenesis and inflammation.

## 1. Introduction

Bone is a hard tissue that supports and shields the body’s other essential organs from harm. Bone is in a state of dynamic equilibrium throughout life, requiring intricate coordination between many types of bone marrow cells. The whole skeleton of an adult human is thought to regenerate every seven years. Continuous bone remodeling is caused by finely controlled mechanisms, including the resorption of bone by osteoclasts and the synthesis of new bone by osteoblasts. While osteoblasts are formed from a shared progenitor cell with adipocytes, bone marrow mesenchymal stem cells (MSCs), osteoclasts develop from hematopoietic stem cell precursors (HSCs) along the myeloid differentiation lineage [1].

Numerous disorders, including osteopetrosis, osteopenia, and osteoporosis, are caused by an imbalance between the production and resorption of bone [2]. These bone abnormalities also play a role in other illnesses like autoimmunity and cancer. The closely regulated lineage commitment of MSCs, a common progenitor, is essential for maintaining bone homeostasis. While MSCs can produce a wide range of cell types, their commitment to osteoblasts and adipocytes has been particularly linked to clinical situations involving aberrant bone remodeling [3]. The study of bone marrow adipose tissue (BMAT) is among the most interesting research subjects of recent years. An increase in BMAT is described in several clinical conditions, in which there is an inverse correlation between BMAT, bone mineral density, and bone quality. For instance, individuals with osteoporosis, the most prevalent bone remodeling disease in the world, have been found to have a higher bone marrow fat content [4,5]. In fact, most bone loss disorders, including aging, and numerous clinical illnesses have been linked to an increase in bone marrow adiposity [6,7,8]. Consequently, adjusting the MSC lineage commitment may offer an efficient treatment plan for associated bone disorders. More thorough research on MSCs’ lineage commitment to adipocytes and osteoblasts is unquestionably necessary, given their shared precursor and the vital functions they perform in the bone marrow microenvironment. Research in these areas will surely shed light on a variety of hematological and metabolic abnormalities that occur in diseases like cancer, osteoporosis, obesity, and aging [3,9]. Recent studies support that, even in anorexia nervosa, BMAT is increased compared to normal weight subjects [3,9].

The aim of this narrative review is to provide an overview of the current research findings on the connection between bone marrow stem cells and obesity, highlighting the multifaceted roles of bone marrow in adipogenesis and inflammation. Therefore, this review summarizes the understanding of the research and clinical uses between bone marrow stem cells and obesity over six years (2016–2022) by searching related keywords in PubMed, Elsevier, and the MDPI database, except for some major references. The search was carried out by matching the following keywords: Mesenchymal stem cell, Bone marrow adipose tissue, Brown adipose tissue, Adipose tissue, Diet, Bone tissue, and Obesity.

## 2. Bone Marrow Mesenchymal Stem Cell Differentiation

Multipotent stem cells, known as hematopoietic stem cells (HSCs) and bone marrow mesenchymal stromal cells (BMSCs), can differentiate into various cell types through intrinsic (such as transcription factors and cofactors, posttranscriptional and posttranslational modifications) and extrinsic (such as secretory molecules, the BM microenvironment, and metabolic cues) regulation [10]. Transcription factors such c-Myc, PU.1/Spi-1, GATA1-3, TNFβ, EGR1, BMI1, Gfi1, FoxO3, and others are involved in coordinating HSCs’ differentiation [11]. For instance, c-Myc controls the ratio of HSCs’ differentiation to self-renewal [12]. Through the regulation of target genes, such as granulocyte colony-stimulating factor receptor [13], granulocyte-macrophage colony-stimulating factor receptor [14], and macrophage colony-stimulating factor receptor, PU.1/Spi-1 plays an important role in the determination of myeloid lineage. Furthermore, a recent investigation showed that, if Notch signaling is diminished, PU.1/Spi-1 expression can drive stem cell differentiation toward the myeloid lineage [15]. Ikaros, a transcription activator that plays a critical role in encouraging lymphocyte differentiation, is another regulating molecule of HSC differentiation. Hypoplasia, the absence of secondary lymphoid organs or the absence of B- and T-cell precursors, is caused by the impairment of this protein [16]. The tumor necrosis factor beta (TNFβ) acts as a negative regulator of HSC self-renewal, whereas the basic leucine zipper transcription factor, ATF-like (BATF), is a crucial element encouraging lymphoid lineage differentiation. Furthermore, B lymphoma Mo-MLV insertion region 1 homolog (BMI1), which is critical for the multilineage potential of HSCs and their capacity for replating, and GATA1-3, zinc finger transcription factors, which coordinate the development of various hematopoietic lineages [17], regulate the determination of the fate of HSC cells [18]. Recently, Lee and colleagues discovered that the control of HSC quiescence and self-renewal is influenced by metabolic status, meaning that it is elevated in obesity and downregulated in weight reduction (Gfi1), a transcriptional repressor. In addition to transcriptional regulation, posttranslational changes, such as DNA methylation, acetylation, or ubiquitination, which can be influenced by aging or metabolic disorders, govern HSCs’ renewal and differentiation [19]. According to recent research, aged HSCs’ B cell lineage output is restored when H4K16Ac levels are raised, because this inhibits Cdc42 [20]. Moreover, elevated H3K9me2 pattern levels linked to HSCs’ lineage commitment are caused by G9a/GLP methyltransferase. However, G9a/GLP suppression reduces stem cell differentiation potential and enhances HSCs’ maintenance [21]. Furthermore, effective hematopoietic differentiation is made possible by DNMT1-mediated methylation [22]. Only a portion of the HSCs’ regulatory network, which collectively demonstrates the intricacy of the stem cell differentiation process, is made up of the transcription factors and posttranslational changes listed above. Certain transcription factors control the differentiation of Bone-Marrow-Derived Mesenchymal Stem Cells (BMSCs) into osteoblasts and adipocytes: Runt-related transcription factor 2 (Runx2), osterix, GATA2 (which determines the lineage of the osteoblasts), peroxisome proliferated-activated receptor gamma (PPARγ) [23], CAAT enhancer binding protein (C/EBP) family [10] (determines the lineage of the adipocytes), Wnt signaling, transforming growth factor β1 (TGF-β1), and bone morphogenic proteins (BMPs) can regulate the activation of these transcription factors [24]. Epigenetic changes also accompany the regulation of BMSCs’ differentiation. For instance, histone deacetylation in genes related to the cellular survival, growth, and proliferation of BMSCs, as well as transcriptional regulation. Runx2, BMP-2, osterix, and osteopontin (OPN) are all expressed more often during osteoblast differentiation and are crucial for osteoblast maturation. The Zfp521, a nuclear protein with 30 Kruppel-like zinc fingers mediating multiple protein–protein interactions, is a crucial regulator of lineage specification in progenitor cells, controlling BMP-induced MSC differentiation in conjunction with histone modification at the Zfp423 promoter, according to a recent work by Addison et al. [25]. These findings show that the differentiation of HSCs and BMSCs is a complex process regulated by distinct transcription factors, the activity of which is further epigenetically modulated; these intrinsic factors are involved in the regulation of BM homeostasis and are influenced by dietary interventions and obesity.

## 3. Adipose Tissue

Adipose tissue (AT) is a very heterogeneous organ because of its multi-depot dispersion in addition to the existence of distinct AT types, such as brown AT (BAT), white AT (WAT), and “brite” (brown-in-white). AT depots have been referred to as “mini organs” with independent traits and functions due to their extreme variability [26]. The BMAT and BAT are two distinct types of adipose tissues with different functions, locations, and characteristics. In Table 1, we report the key differences between them (Table 1).

A recent study of BAT in adults demonstrated the presence of unilocular adipocytes that are positive for mitochondrial brown fat uncoupling protein 1 (UCP1), and the stromal vascular fraction of adipose biopsies in this region contained preadipocytes that differentiated into thermogenic adipocytes in vitro [27]. Overall, these findings highlight the plasticity of BAT and appear to reflect a continuous environment-dependent adaptation of this tissue. Although sympathetic activity is probably an important general regulator of this plasticity, downstream mediators, i.e., “batokines”, which act specifically on the different cell types that make up BAT, could be valuable tools to increase the quantity of brown adipocytes that are mature and active in adult humans. A crucial trait of AT, in addition to its variety, is its plasticity—the capacity to alter its structural, cellular, and molecular features in response to pathological and physiological circumstances that directly affect its activity [27]. The impact of this flexibility on AT functionality determines whether it is harmful or advantageous. Furthermore, AT plasticity and remodeling are also influenced by structural and cellular variations among the various fat depots. The two main depots of white adipose tissue (WAT)—subcutaneous adipose tissue (SAT), commonly considered as an energy storage organ, and visceral adipose tissue (VAT), traditionally believed to provide protection against trauma—exhibit notable differences. Their adipokine expression profiles, vascular density, and innervation vary, leading to distinct metabolic roles [28].

Crucially, VAT is regarded as a more harmful WAT depot than SAT because it has a stronger acute inflammatory profile [29].

As a result, AT is an incredibly dynamic organ whose properties can vary greatly based on regulatory and conditioning variables. A strong decrease in brown adipocytes and a corresponding rise in white adipocytes with aging are two of the most noticeable and frequent physiological changes seen in AT [30]. Remarkably, the opposite effect appears to occur following exposure to cold in both humans and rodents [31], providing a glaring illustration of the possible advantageous adaptability of AT. In a similar vein, exercise can encourage AT browning [32]. Furthermore, from a functional point of view, data in the literature reported that cold exposure does not stimulate BMAT glucose uptake in mice or humans, with mouse BMAT also showing no induction of the BAT and beige markers Ucp1, Dio2, Prkaa1, and Metrnl. In addition, some studies suggest that cold exposure may have variable effects on BMAT. Some studies propose that cold exposure could lead to a reduction in BMAT, as the body prioritizes the activation of BAT for thermogenesis. BAT activation is associated with increased energy expenditure, and, in some cases, this may result in a decrease in overall adiposity, including BMAT [30,31]. However, the specific impact of cold on BMAT can depend on various factors, including the duration and intensity of cold exposure, individual differences, and the specific mechanisms involved. While there is evidence suggesting a relationship between temperature exposure and adipose tissue dynamics, more research is needed to fully understand the nuances of how cold affects BMAT [32,33].

The presence of cardiovascular risk factors, in particular obesity, is one of the most important factors causing a metabolic shift in adipocytes that results in dysfunctional AT, one of the few pathological conditions where AT plasticity is found. BAT atrophy is observed in obese people along with elevated fat accumulation in VAT and hyperglycemia [33].

Increased expression of pro-inflammatory cytokines, such as TNFα, IL1, IL-6, and the receptor activator of RANKL/OPG/RANK system, which controls osteoclast development, is linked to estrogen deficit in the bone microenvironment [34,35,36]. Reduced endogenous antioxidant production is another effect of estrogen deficiency [37]. When combined with the increased production of reactive oxygen species (ROS) and the senescence microenvironment seen in aging humans and obese mice, these factors can cause a marked increase in oxidative stress in the bone microenvironment [34,38]. Increased bone turnover in response to estrogen deficiency was found in studies involving postmenopausal women and ovariectomized rats. This increased bone resorption and formation was demonstrated by an increase in osteoblast precursors, osteoblast proliferation, and osteoblast number [39,40]. But bone loss results from heightened bone resorption, which is not sufficiently compensated for by increasing levels of bone production. An interesting question that has been raised is whether biological sex has an impact on the expansion of BMAT during aging. Griffith et al. investigated BMD and BMAT in the lumbar spine of 259 healthy subjects (145 females and 114 males; age range: 62–90 years) using MR spectroscopy of the L3 vertebral body, and revealed that, in males, BMAT increased gradually throughout life, whereas in females, BMAT increased between 55 and 65 years [41]. BMAT in the vertebral bones in females with an age of more than 60 years increased by nearly 10% higher compared to age-matched males, indicating the positive impact of estrogen deficiency on BMAT expansion. Moreover, research on humans has demonstrated that, while osteoporosis in men significantly increases vertebral BMAT [42], aging and the loss of steroid hormones make women more susceptible to the detrimental effects of BMAT on trabecular bone loss at the spine and femoral neck, as well as a greater loss of spine strength [43]. A review of various clinical and preclinical investigations on the relationship between BMAT and endocrine aging is given.

To develop targeted strategies to counteract the presence of dysfunctional AT, a deeper understanding of the effects of individual-related factors, such as gender or age, and the clustering of cardiovascular risk factors, such as hypercholesterolemia, hyperglycemia, and obesity-associated subclinical inflammation, on AT heterogeneity and plasticity is essential.

## 4. AT Pathophysiology

It has long been believed that adipose tissue is a passive energy source. But since the discovery of adipokines like leptin, this idea has evolved, with adipose tissue now being recognized as an endocrine organ that plays a crucial role in maintaining energy balance. There are several possible molecular routes via which adipose tissue and bone interact. This dynamic and active interaction involves several different elements, including pro-inflammatory cytokines, adiponectin, leptin, and vitamin D. Additionally, through bone-derived substances like osteocalcin and osteopontin, the bone tissue influences metabolic parameters, including body weight regulation. Adipose tissue and bone are generally associated in two ways: mechanically and metabolically [44]. Sarcopenic obesity refers to a condition characterized by the coexistence of sarcopenia (loss of muscle mass) and obesity (excess body fat). This combination can have various effects on the body and the interplay between sarcopenic obesity and BAT. Sarcopenic obesity has been associated with changes in adipose tissue distribution, including an increase in visceral adiposity. It is plausible that these changes may extend to the bone marrow, potentially leading to an increase in BMAT content. Sarcopenic obesity is often linked to chronic low-grade inflammation and metabolic disturbances. These factors can influence the bone marrow microenvironment, potentially affecting the composition and function of BMAT. Furthermore, obesity and sarcopenia can both influence hormonal balance. Changes in hormones such as leptin, adiponectin, and inflammatory cytokines may have downstream effects on BMAT and bone metabolism [45].

### 4.1. Mechanically

Compared to lean individuals, obese individuals have lower levels of biochemical indicators of bone turnover [44]. When it comes to bone resorption markers, this distinction appears to be more significant than it is for bone production markers. These effects support the maintenance of bone mass in maturity. It appears that a greater body weight slows down bone loss during menopause [46].

The increased mechanical loading and strain associated with obesity is the most likely mechanism to explain the elevated bone mineral density in obese subjects.

Subjects affected by obesity have a higher body fat mass in addition to a higher lean mass, with very few exceptions (such as severe sarcopenia). This has positive impacts on bone geometry and modeling, in addition to increasing muscular strain and passive loading [47]. An increase in bone size due to bone apposition should be anticipated if physical loading is the only factor causing the elevated BMD. Results, however, do not always support this theory: high-resolution peripheral quantitative computed tomography (CT) estimates of bone size at the radius and tibia show no differences between obese and normal-weight controls [46,48]. These results indicate that, while the loading factor contributes to the bone–fat relationship, it is not enough to fully explain the interaction. Overweight has been linked to falls in several studies conducted in the past few years, particularly among older adults. This high rate of falls results in significant medical and financial expenses [48]. The cause of falls in individuals who are overweight or obese is complex. First, chronic health issues like diabetes, hypertension, arthritis, hypoventilation syndrome, sleep apnea, and cardiovascular disease can all be brought on by or made worse by obesity [49]. Peripheral neuropathy, autonomic dysfunction with orthostatic hypotension and instability, and overall weakness are all highly correlated with these disorders and are risk factors for falls [49]. Second, carrying too much weight is associated with a decreased capacity to perform daily activities like walking alone or ascending stairs, which raises the chance of falling once more [50]. Third, carrying more weight puts more strain on the heels, impairing balance and postural stability [51]. Fourth, since it undermines the stability of the body center, central adiposity in older women, as determined by the waist-to-hip ratio, has a significant independent function as a fall-related indicator [52]. New clinical entities have been identified in the past few years to clarify the intricate physiological relationship between falls and obesity. The hallmark of “dynapenic obesity” is the loss of muscular mass brought on by obesity. It increases the risk of falls and is linked to restricted mobility [53]. The hallmark of “sarcopenic obesity” is the loss of muscular mass brought on by fat. Because of postural instability and decreased physical activity, sarcopenia is positively correlated with an increased risk of falls, as well as a loss of bone mineral density and osteoporosis, which increases the risk of fractures in older persons [53,54,55,56]. The site-specificity of fracture risk is influenced by the reduction in preventive mechanisms and the falling pattern seen in obese people [57]. Women who are obese are less likely to break their hips, but they are still at a significant risk of breaking other bones. The existence of adipose tissue (padding) around the femur and pelvis, which lessens the force of falling, could be one explanation for the low incidence of hip fractures [58]. When comparing obese and overweight people to lean people, they tend to fall in different patterns: the former are more likely to fall sideways or backward, while the latter are more likely to fall forward [59]. The greater frequency of fractures at the ankle and lower leg locations in obese people may also be due to exaggerated introversion and extroversion of these regions. More fractures in the ankle, leg, humerus, and vertebral column and fewer in the wrist, hip, and pelvis occur in obese women. In conclusion, the evidence currently available indicates that obesity raises the risk of falls and multiple falls in individuals over 60, who are at a higher risk of experiencing lasting disability and having a lower quality of life [59,60,61,62,63]. On the other hand, the evidence linking obesity to fall-related injuries or fractures is weaker.

### 4.2. Metabolic

One of the main sources of aromatase, an enzyme that synthesizes estrogens from androgen precursors and is also expressed in the gonads, is adipose tissue. Estrogens are steroid hormones that are essential for maintaining skeletal homeostasis because they protect bone by increasing bone production and decreasing bone resorption. Research has demonstrated that the serum concentrations of estrogens are higher in obese post-menopausal women than in non-obese controls [37]. These results could help to explain why BMD and BMI have a positive correlation. It is now clear, though, that estrogens are not the only hormone that controls bone mass. The ob (Lep) gene produces the protein leptin. It is a hormone that resembles a cytokine and is mostly made by adipocytes. It primarily regulates appetite and energy homeostasis by causing satiety in the hypothalamus. Being obese, a leptin-resistant condition, usually results in higher leptin concentrations [38]. Due to insulin resistance and hyperinsulinemia, hyperleptinemia is a separate risk factor for cardiovascular disease [39]. Leptin has complicated effects on bone [40,41]. It affects people directly as well as indirectly through peripheral and central hypothalamic pathways. There have been reports of both beneficial and detrimental effects on BMD, and the outcomes of in vivo research have been inconsistent. While mature osteoclasts appear to be unaffected, leptin enhances the differentiation of stromal cells to osteoblasts in vitro [42], boosts osteoblast proliferation, and suppresses osteoclast genesis [43]. Moreover, a decrease in bone volume and BMD results from a defect in leptin signaling brought on by a deletion of the receptor gene for the hormone [44]. Studies conducted in vivo indicate that the location and method of action of leptin may affect its effects. While central leptin administration—as an intracerebroventricular infusion—induces bone loss in both leptin-deficient and wild-type mice, it has been suggested that peripheral leptin administration increases bone mass by promoting bone formation and inhibiting bone resorption [45,46]. The sympathetic nervous system appears to be the mechanism causing this detrimental central effect. The hypothalamic neuropeptide Y (NPY), which is necessary for controlling energy and food intake, as well as bone remodeling, is suppressed by leptin. Even more contradicting data come from human sources, maybe because of study restrictions. Research has indicated that leptin has both fundamentally harmful functions [50,51] and favorable ones [48,49]. In conclusion, it does not appear plausible that the hyperleptinemia seen in obese people is bad for the bone.

Insulin resistance, diabetes, and obesity are all associated with low adiponectin concentrations [64,65,66]. They function as protective factors in the cardiovascular system and glucose metabolism, and they become better after losing weight [53]. Studies conducted both in vivo and in vitro indicate that adiponectin promotes osteoblast genesis and inhibits osteoclast genesis, which is beneficial for bone mass [54]. These results suggest indirectly that BMD may be enhanced by increased adiponectin concentrations after weight loss and fat reduction. Conversely, there is an inverse relationship between the amounts of TNF-α, IL-6, and C-reactive protein (CRP) and adiponectin. Strong inhibitors of adiponectin expression include these markers of inflammation [55]. This may suggest indirectly that bone quality may be negatively impacted by long-term inflammatory processes, such as central and visceral obesity. Pro-inflammatory cytokines like TNF-α and IL-6 are linked to obesity. In humans, the percentages of body fat and insulin resistance are correlated with the expression of TNF-α [56]. By increasing the amounts of c-fms, receptor activator of nuclear factor kappa-Β ligand (RANK), and RANK ligand (RANKL), it causes bone loss by osteoclast genesis through the activation of NFκB in obesity [57]. The main osteoclastogenic cytokine factor that stimulates osteoclasts’ resorptive activity is RANKL [58,59]. Moreover, TNF-α decreases the synthesis of osteoprotogerin (OPG), an inhibitor of RANKL, which raises RANKL concentrations and causes additional bone loss [60]. Lastly, TNF-α directly affects the signaling pathways that RANKL induces, resulting in a synergistic interaction with RANKL that encourages more osteoclastic resorption [61]. It appears that this characteristic sets the cytokines apart. Another cytokine with a variety of functions is IL-6. Adipocytes and fibroblasts are among the cells that upregulate its production [62] in response to obesity and insulin resistance [63]. Like TNFα, IL-6 stimulates bone resorption and osteoclast genesis. Moreover, pre-osteoblast differentiation and osteoblast proliferation are stimulated by IL-6 [64]. According to new research, inflammatory cytokines may be a major factor in bone loss [65]. Low-grade chronic inflammation is linked to obesity; this inflammation is particularly prominent in central and visceral adiposity, which is defined by elevated amounts of CRP, TNF-α, and IL-6. The rapid bone loss seen in obesity may be caused by this strong inflammatory response.

It is common for obese people to be deficient in vitamin D. It is commonly known how low vitamin D levels affect the musculoskeletal system. Serum 25(OH)D is inversely correlated with body weight, BMI, and fat mass in obese individuals, and obese individuals have lower serum 25(OH)D concentrations than normal-weight individuals. Obese individuals have serum 25(OH)D concentrations that are about 20% lower than those of normal-weight individuals [67]. Compared to that of normal-weight individuals, the prevalence of vitamin D in obese people is between 40 and 80% higher [49]. Surprisingly, compared to obese adults and children, non-obese adults and children appear to benefit from vitamin D supplements much more in terms of raising vitamin D concentrations [67]. However, a meta-analysis revealed no change in body weight or fat mass [67]. The sequestration of vitamin D in fat storage is probably the reason why vitamin D concentrations are restored more successfully in normal-weight people than in obese people. Potentially, secondary hyperparathyroidism is another issue. Secondary hyperparathyroidism affects up to 43% of morbidly obese people, which has an adverse effect on skeletal health [38]. Low total 25(OH) concentrations in various clinical scenarios would result in higher bone turnover, decreased BMD, and decreased calcium absorption from the diet. On the other hand, compared to people of normal weight, obese individuals appear to have decreased bone turnover and higher BMD due to thicker and denser cortices [13]. On the other hand, obesity has a detrimental effect on bone strength in children and adolescents, which is another concerning effect of juvenile obesity [60]. The reasons why vitamin D insufficiency in obese individuals does not have a detrimental effect on bone are not fully understood. Adults who are obese and deficient in vitamin D may, according to one theory, develop compensatory mechanisms to offset the detrimental effects of vitamin D [57]. These systems involve elements including estrogens, mechanical stress, and leptin. Another idea holds that people who are obese do not actually have low levels of vitamin D. According to this theory, the reservoir in fat tissue maintains a sufficient supply of vitamin D, even while serum 25(OH)D concentrations are lower. This results in higher total body vitamin D levels.

An important investigation proposed that the vitamin D receptor (VDR) inhibits adipogenesis through direct inhibition of the expression of PPARγ, the master transcriptional regulator of the adipogenic gene program [68]. The PPARγ agonist troglitazone reversed the inhibition of adipogenesis by calcitriol, and conversely, calcitriol antagonized the transactivation capacity of PPARγ in 3T3-L1 cells [68]. VDR overexpression experiments showed that even unliganded VDR could inhibit PPARγ transactivation activity, by competing for binding to their common heterodimer partner RXR [68]. Numerous studies have also demonstrated the complementary effects of PPARγ2 and calcitriol in bone marrow and BMSCs, which are consistent with this mechanism. In human bone marrow BMSCs, calcitriol decreased the expression of PPARγ2 [69]. In contrast, the PPARγ antagonist BADGE increased the expression of VDR in bone, enabling calcitriol to enhance the anti-adipogenic effects of BADGE in bone marrow [70]. The expression of PPARγ was upregulated in cultured bone marrow stem cells (BMSCs) derived from VDR null mice. Additionally, there was an increase in adipogenesis and higher expression levels of DKK1 and SFRP2, which are inhibitors of the pro-osteogenic canonical Wnt signaling pathway. In wild-type BMSCs, calcitriol was shown to downregulate the expression of these Wnt inhibitors in both the absence and presence of adipogenic inducers [55]. The selection between osteoblast genesis and adipogenesis is regulated by the reciprocal inhibition of PPARγ2 and the Wnt pathway [71]. Therefore, vitamin D may restrict BMAT growth through the inhibition of PPARγ2 and the promotion of Wnt signaling.

### 4.3. Diet and Bone Tissue

Obesity brought on by a high-fat diet (HFD) is likely the most often used model to examine the effects of obesity on bone metabolism. Animal model data point to a detrimental impact of obesity on bone metabolism. Both systemic inflammation and changes in the bone microenvironment most likely contribute to this detrimental effect. One important discovery was that obesity brought on by a HFD was linked to higher bone quantity (bigger size and mineral content) but inferior bone quality (lower size-independent mechanical properties) [59]. According to a different study, mice given a high-fat diet experienced a significant and early-stage deterioration in the micro-architecture of their trabecular bones, which can ultimately result in a decrease in trabecular bone density. Furthermore, increased bone resorption and increased bone marrow adiposity are caused by obesity generated by a high-fat diet. Bone resorption appears to be linked to the bone microenvironment in obesity [60,64,67]. A recent study reported that HFD-induced obesity in mice led to increased BMAT formation, enhanced AD differentiation of BM-MSCs, and reduced bone mass. Interestingly, BMAT, in contrast to extramedullary adipocytes, did not exhibit a pro-inflammatory phenotype and maintained its insulin responsiveness [38]. This study suggests that BMAT responds to metabolic stimuli and acts as a depot for storing extra energy during a HFD, when peripheral AT is incapable of mediating this function. A different study observed that a HFD increased BMAT formation, as evidenced by an increased BMAT volume, adipocyte number, and size [72,73]. Other investigations have reported inconsistent results regarding the impact of a HFD on bone mass and bone quality. Some studies have reported no change or increased or decreased bone mass with a lower bone strength [74,75]. These discrepancies can be explained by variations in experimental models, e.g., mouse strain, sex, length/type/composition of diet (60% versus 45% fat or 10% corn oil), and possibly evaluation methods. However, a recent study by Bornstein and colleagues reported that a HFD induced a reduction in the cortical and trabecular parameters in the long bones, along with an increased BMAT volume, suggesting that bone mass reduction is a biological consequence of the coherent mechanisms of cellular and molecular changes relevant for studies of bone fragility in human obesity [38].

## 5. Obesity-Induced Inflammation

Studies using human and animal models have demonstrated that obesity is associated with a considerable increase in macrophage infiltration. This recruitment of macrophages is connected to oxidative stress, insulin resistance, and systemic inflammation [76]. Several studies have shown a close association between adipose tissue and lymphatic tumors, probably supported by the secretion of adipokines, cytokines that can regulate the proliferation, activation, and secretory activity of different immune cells. In fact, much research has suggested the involvement of adipokines in the defense mechanisms of the immune system and in inflammatory processes.

AT and inflammation are closely associated, not only because obesity is associated with chronic low-grade inflammation [77], but also because adipocytes naturally secrete both pro- and anti-inflammatory adipokines [78], with the imbalance between these adipokines determining the AT functional profile and the emergence of metabolic disorders. Greater in size (hypertrophic) than adipocytes from lean subjects, obese individuals’ adipocytes alter their adipokine secretion profile toward a pro-inflammatory state, resulting in lower levels of adiponectin but higher levels of tumor necrosis factor-alpha (TNF-alpha). This pro-inflammatory milieu is thus linked to obesity. The inflammation caused by fat has a direct effect on AT performance [79].

Adipogenesis, the process by which stem cells differentiate into adipocytes, is one reason why a high calorie intake induces adipocyte hyperplasia in addition to adipocyte hypertrophy [80]. On the one hand, freshly created adipocytes are more insulin-sensitive and take in more triglycerides and free fatty acids [81]. Conversely, hypertrophied adipocytes degenerate into defective cells that accumulate fat ectopically and lose their capacity to guard against systemic lipotoxicity. The remodeling of the extracellular matrix to facilitate tissue expansion is another feature of obesity [82]. Furthermore, AT from obese individuals demonstrates a reduced capacity to enlarge the capillary network enveloping adipocytes, resulting in adipocyte hypoxia and necrosis [83,84].

Therefore, the appearance of crown-like structures (CLS), which are created by invading macrophages surrounding necrotic adipocytes, is indicative of obese AT [85]. However, the background characteristics of each AT depot determine how obesity and the inflammatory condition that goes along with it affect AT development differently.

Not only can AT generate and secrete cytokines and adipokines, but they can also form extracellular vesicles (EVs). All EVs have a similar composition to their original cells and are loaded with bioactive substances like proteins, lipids, and DNA that are transported to cells in distant organs or within ATs to facilitate communication between cells within and between organs. In this regard, it has been determined that AT-derived extracellular vesicles (ADEVs) are essential for immunological and metabolic response cellular communication, controlling cellular functions in nearby and distant tissues. This study will concentrate on the characteristics and roles of ADEVs from various cellular sources in AT, as well as their role in maintaining AT homeostasis and preventing the emergence of metabolic problems, including metabolic disorders, cardiovascular disease (CVDs), various types of cancer, and neurological disorders [86]. EVs are small membrane-bound particles released by cells, and they play a crucial role in intercellular communication. These vesicles can carry various bioactive molecules, such as proteins, lipids, and nucleic acids. Studies have investigated extracellular vesicles derived from bone marrow and adipose tissue for their potential roles in communication between cells and tissues. Bone Marrow Adipose Tissue, being a specialized type of adipose tissue located within the bone marrow, is known to influence various physiological processes, including hematopoiesis and skeletal homeostasis [87].

## 6. Obesity and Bone Marrow Adipose Tissue

The connection between BMAT and bone homeostasis has been thoroughly studied using animal models [88,89,90,91]. In models of obesity generated by diet, BMAT constantly expands [75,78,92,93]. Remarkably, skeletal alterations have been linked to obesity [94]. It is not unexpected that BMAT increases in mouse models of high-fat diets (HFD) due to the lipid accumulation in all adipocytes through various fat depots [75,78,92,93]. Even though BMAT expansion has been shown to occur consistently in HFD animal models, different effects on bone have been reported [75,78,92,93]. These variations in results are likely due to variations in the experimental conditions, such as the type of diet used, the age at which HFD feeding began, and the species, strain, and gender of the selected animal [95]. Exercise has been shown to improve bone integrity and decrease BMAT growth in obese individuals on a high-fat diet. Skeletal alterations linked to obesity are more reliable in people. Because people living with obesity (PwO) exhibit normal or even higher BMD due to mechanical adaptations to increased body weight, obesity has long been considered to be a preventive factor against osteoporosis [95]. Nevertheless, this assumption has been contested by several studies [88,96,97]. PwO have deficiencies in the quality of their bone matrix and structure, lower levels of bone remodeling, and increased fracture risk in unique skeletal locations (proximal humerus, ankle, and upper leg). Hormonal imbalances associated with elevated (central) obesity and chronic inflammation are mostly responsible for these latter effects on bone metabolism [88,96]. There are not many clinical studies assessing BMAT in PwO and its relationship with bone metabolism, but it is possible that BMAT plays a role in some of the worsening components linked to bone quality deterioration in obesity [88]. Cohen et al. assessed bone micro-architecture, remodeling, and BMAT in healthy premenopausal women of different weight ranges using histomorphometry in labeled transiliac bone samples [98,99]. They also assessed the BMD and trunk fat (a proxy for visceral adipose tissue, or VAT) in those forty premenopausal women (37.3 ± 8.2 years) with BMIs ranging from 20.1 to 39.2 kg/m^2^. The measurements were made using Dual-Energy X-ray Absorptiometry (DEXA). Individuals in the highest tertile of trunk fat (*n* = 13, BMI 29.8 ± 4.6 kg/m^2^) had significantly poorer bone formation and inferior bone quality (lower trabecular bone volume and higher cortical porosity) in comparison to those in the lowest tertile (*n* = 13, BMI 21.6 ± 1.3 kg/m^2^). Adipocyte number (#/mm^2^) 165.9 ± 39.8 versus 164.6 ± 35.9 (*p*-value adjusted for age = 0.30) and adipocyte volume/marrow volume (%) 27.9 ± 8.1 versus 22.6 ± 5.5 (*p*-value adjusted for age = 0.40) were similar in women in the highest tertile of trunk fat, according to transiliac crest bone biopsies. Regression studies revealed a clear correlation between bone marrow adiposity (BMA) (adipocyte volume/marrow volume) and trunk fat (r = 0.37, *p* = 0.019). BMA tended to be larger in the highest tertile and was considerably higher when the two upper tertiles were combined. In addition, they postulated that insulin-like growth factor 1 (IGF-I), as previously proposed by Bredella et al. [99], may operate as a mediator in the interaction between fat (trunk fat and BMA) and bone. In a different study conducted by Bredella et al. [100], 1H-MRS of the L4 vertebrae was performed on 106 healthy young men and women (mean age, 33.7 years ± 6.8; mean BMI, 33.1 kg/m^2^ ± 7.1). Examining the relationships between ectopic lipid levels, serum lipid levels, and BMAT was the major goal. The interpretation of this study is limited, because the bone parameters were not examined. Despite this restriction, PwO (*n* = 88, 0.75 ± 0.38) and normal-weight controls (*n* = 18, 0.57 ± 0.20) did not vary substantially (*p* = 0.07) in L4 vertebral BMAT (lipid/water ratio), but BMAT tended to be greater in PwO. In conclusion, there are few clinical trials assessing BMAT in PwO, and the findings are inconsistent. The differences in the study population (premenopausal women, young men and women, and adolescent girls), the methods used to measure BMAT (1H-MRS and histomorphometry), and the measurement site (vertebrae and femoral diaphysis) may have contributed to the inconsistent results when comparing the BMAT levels in PwO with those of normal-weight controls. On the other hand, patients with the highest VAT tended to have higher BMA, and inverse relationships between BMA and BMD were repeatedly seen. This shows that BMA may have an impact on bone health at both a local and systemic level, necessitating more research.

## 7. Conclusions

Adipose tissue and bone have a complicated relationship. Adipokines, estrogens, and metabolic substances derived from bone facilitate the interaction between the two tissues, which have very active metabolisms. Feedback processes that impact bone remodeling, body weight regulation, adipogenesis, glucose homeostasis, and muscle adaptation complicate the interplay between them. As predicted, mechanical loading improves bone health; however, in obese people, this may not be enough. Growing evidence points to a detrimental effect of obesity on bone health. Because obesity is associated with an up-regulation of pro-inflammatory cytokines and/or increased production of leptin, low-grade systemic inflammation is likely detrimental to bone health. A further possible explanation is that obese people’s increased bone marrow adipogenesis could result in a lower bone mass, since obesity is linked to an abnormal commitment of the common progenitor stem cell into adipocytes rather than osteoblasts.

Research is currently continuing to better understand the complex physiology of bone marrow adipose tissue; certainly, physical exercise combined with correct nutrition represents “a winning choice” to combat the deleterious effects of obesity and its complications on bone health. Practicing physical activity brings numerous health benefits and is essential for strong bones and muscles. An increase in physical activity levels also determines an improvement in body composition and a more favorable biochemical profile in terms of cardiovascular prevention, diabetes mellitus, and bone pathologies, and better cardiorespiratory fitness (CRF), which, in addition to being a reliable measure of regular physical activity, is also an important indicator of health status. The combination of an active life, of a diet rich in calcium, and the right intake of vitamin D ensures the opportunity to improve the health of the skeleton and muscles and reduce the risk of osteoporosis. In conclusion, the study of the interaction between bone marrow stem cells and obesity represents a future challenge for the identification of new therapeutic targets for multiple chronic diseases.

## Figures and Tables

**Table 1 ijms-25-02715-t001:** The main differences between bone marrow adipose tissue (BMAT) and brown adipose tissue (BAT).

Key Differences	BAT	BMAT
**Function**	The primary function of BMAT is not entirely clear, but it is believed to play a role in energy storage and bone metabolism. BMAT can also influence bone health and hematopoiesis, the process of blood cell formation that occurs in the bone marrow.	BAT is specialized for thermogenesis, a process that generates heat in response to cold- or diet-induced factors. It contains a high number of mitochondria and expresses uncoupling protein 1 (UCP1), which uncouples the respiratory chain to produce heat instead of ATP.
**Location**	Found within the bone marrow cavities of long bones and in other skeletal locations. It is distributed throughout the skeleton.	Typically located in discrete depots, such as the interscapular region, neck, and around major blood vessels. BAT has a more limited distribution compared to BMAT.
**Cell type**	Contains adipocytes, which are cells specialized for fat storage. The differentiation of mesenchymal stem cells in the bone marrow into adipocytes contributes to the formation of BMAT.	Composed of brown adipocytes, which have a unique appearance due to the presence of numerous mitochondria and a high vascularization. These characteristics are essential for BAT’s thermogenic function.
**Color and apparence**	Appears yellowish and resembles white adipose tissue. Its color is due to the accumulation of triglycerides in adipocytes.	Appears brown due to the high concentration of mitochondria and the presence of iron-containing cytochromes. The brown color is a result of increased vascularity and mitochondrial density.
**Physiological role**	Implicated in bone health, hematopoiesis, and potentially providing a local energy source for bone cells. The exact physiological significance of BMAT is still an active area of research.	Primarily involved in non-shivering thermogenesis, helping to maintain body temperature in response to cold exposure. BAT activity can also impact energy expenditure and metabolic homeostasis.
**Association with health conditions**	Increased BMAT is often associated with conditions such as obesity and certain metabolic disorders. The relationship between BMAT and overall health is complex and continues to be explored.	BAT activation is associated with improved metabolic health and has been investigated as a potential target for treating obesity and related metabolic disorders.

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
