# Peer review of "Exploring the Interplay between Bone Marrow Stem Cells and Obesity"

_ijms, 2024, doi:10.3390/ijms25052715_

Round 1

Reviewer 1 Report

Comments and Suggestions for Authors

Review comments:

The topic of the review is interesting; however, the authors focused on Obesity and failed to draw a relationship between increased Obesity and BMAT. There is no statistic mentioned.

Comments:

Abstract: Correct “Mesenchimal” to Mesenchymal

Introduction:

Figure 1. Please provide detailed figure legends.

All factors such as TNFβ should be spelled out first in the text.

Page 4:

1.      Please define “perirenal BAT” and how is it different from BMAT.

2.       Line 155. The sentence is not clear. Is it increasing or decreasing?  

3.      What is the effect of cold on BMAT?

Page 5:

1.      Line 185: Does menopause affect BMAT cell number, proliferation, or differentiation? Are there any published studies?

2.      What is BMD?

3.      How does sarcopenic Obesity affect BMAT?

Page 7:

1.      Are there any studies linking low vitamin D with a decrease in BMAT differentiation?

2.      Please remove number 71 from the text

3.      Does high-fat diet result in high BMAT? Is there any effect of bone resorption on the isolation of BMAT?

Page 8:

1.      Are there any studies showing EVs formed from BMAT? Are they similar to EVs isolated from AT?

Comments on the Quality of English Language

none

Author Response

Review 1 comments:

The topic of the review is interesting; however, the authors focused on Obesity and failed to draw a relationship between increased Obesity and BMAT. There is no statistic mentioned.

Reply

Dear reviewer, thank you for your valuable suggestions which will be fundamental to improving our manuscript. We agree with you that we need an explanation that better highlight the relationship between increased Obesity and BMAT. For these reasons we have decided to include a paragraph that explains these topics in detail. Below we report the section included in our manuscript:

“ 6. Obesity and bone marrow adipose tissue

The connection between BMAT and bone homeostasis has been thoroughly studied using animal models [70–73]. In models of obesity generated by diet, BMAT constantly expands [74–77]. Remarkably, skeletal alterations have been linked to obesity [78]. It is not unexpected that BMAT increases in mouse models of high-fat diets (HFD) due to the lipid accumulation in all adipocytes through the various fat depots [74–77]. Even though BMAT expansion has been shown to occur consistently in HFD animal models, different effects on bone have been reported [74–77]. These variations in results are likely due to variations in the experimental conditions, such as the type of diet used, the age at which HFD feeding begins, and the species, strain, and gender of the selected animal [79]. Exercise has been shown to improve bone integrity and decrease BMAT growth in obese individuals on a high-fat diet. Skeletal alterations linked to obesity are more reliable in people. Because people living with obesity (PwO) exhibit normal or even higher BMD due to mechanical adaptations to increased body weight, obesity has long been considered a preventive factor against osteoporosis [79]. Nevertheless, this assumption has been contested by several research [70,80,81]. PwO have deficiencies in the quality of the bone matrix and structure, lower levels of bone remodeling, and increased fracture risk in unique skeletal locations (proximal humerus, ankle, and upper leg). Hormonal imbalances associated with elevated (central) obesity and chronic inflammation are mostly responsible for these latter effects on bone metabolism [70,80]. There aren't much clinical research assessing BMAT in PwO and its relationship to bone metabolism, but it's possible that BMAT plays a role in some of the worsening components linked to bone quality deterioration in obesity [70]. Cohen et al. assessed bone microarchitecture, remodeling, and BMAT in healthy premenopausal women of different weight ranges using histomorphometry in labeled transiliac bone samples [82,83]. They also assessed the BMD and trunk fat (a proxy for visceral adipose tissue, or VAT) in those forty premenopausal women (37.3 ± 8.2 years) with BMIs ranging from 20.1 to 39.2 kg/m2. The measurements were made using Dual-Energy X-ray Absorptiometry (DEXA). Individuals in the highest tertile of trunk fat (n = 13, BMI 29.8 ± 4.6 kg/m2) had significantly poorer bone formation and inferior bone quality (lower trabecular bone volume and higher cortical porosity) in comparison to those in the lowest tertile (n = 13, BMI 21.6 ± 1.3 kg/m2). Adipocyte number (#/mm2) 165.9 ± 39.8 versus 164.6 ± 35.9 (p-value adjusted for age = 0.30) and adipocyte volume/marrow volume (%) 27.9 ± 8.1 versus 22.6 ± 5.5 (p-value adjusted for age = 0.40) were similar in women in the highest tertile of trunk fat, according to transiliac crest bone biopsies. Regression studies revealed a clear correlation between bone marrow adiposity (BMA) (adipocyte volume/marrow volume) and trunk fat (r = 0.37, p = 0.019). BMA tended to be larger in the highest tertile and was considerably higher when the two upper tertiles were combined. In addition, they postulated that insulin-like growth factor 1 (IGF-I), as previously proposed by Bredella et al.  [83], may operate as a mediator in the interaction between fat (trunk fat and BMA) and bone. In a different study conducted by Bredella et al. [84], 1H-MRS of the L4 vertebrae was done on 106 healthy young men and women (mean age, 33.7 years ± 6.8; mean BMI, 33.1 kg/m2 ± 7.1). Examining the relationships between ectopic lipid levels, serum lipid levels, and BMAT was the major goal. The interpretation of this study is limited because the bone parameters were not examined. Despite this restriction, PwO (n = 88, 0.75 ± 0.38) and normal-weight controls (n = 18, 0.57 ± 0.20) did not vary substantially (p = 0.07) in L4 vertebral BMAT (lipid/water ratio), but BMAT tended to be greater in PwO. In conclusion, there are few clinical trials assessing BMAT in PwO, and the findings are inconsistent. The differences in the study population (premenopausal women, young men and women, and adolescent girls), the methods used to measure BMAT (1H-MRS and histomorphometry), and the measurement site (vertebrae and femoral diaphysis) may have contributed to the inconsistent results when comparing BMAT levels in PwO with normal-weight controls. On the other hand, patients with the highest VAT tended to have higher BMA, and inverse relationships between BMA and BMD have been repeatedly seen. This shows that BMA may have an impact on bone health on both a local and systemic level, necessitating more research.”

Review 1 comments:

Comments: 

Abstract: Correct “Mesenchimal” to Mesenchymal

Reply

We changed the word following your suggestion.

Review 1 comments:

Introduction: 

Figure 1. Please provide detailed figure legends. 

Reply

Following your suggestion we modified the Figure 1 caption as follow:

Figure 1. The figure shows the Multifaceted Connections of Mesenchymal Stem Cells - Unveiling Their Regenerative, Immunomodulatory, and Tissue Engineering Capabilities in Biomedical Research. (MSC: mesenchymal stem cells).”

Review 1 comments

All factors such as TNFβ should be spelled out first in the text. 

Reply

We specified the acronym in the text.

Review 1 comments

Page 4: 

Please define “perirenal BAT” and how is it different from BMAT.

Reply

Dear reviewer, thank you for your valuable suggestions which will be fundamental to improving our manuscript. We agree with you that we need an explanation that better highlight the differences between BAT and BMAT. To better explain this section, we added a table:

Table 1. In this table were reported the differences between Bone marrow adipose tissue (BMAT) and brown adipose tissue (BAT).

Key differences

BAT

BMAT

Function

The primary function of BMAT is not entirely clear, but it is believed to play a role in energy storage and bone metabolism. BMAT can also influence bone health and hematopoiesis, the process of blood cell formation that occurs in the bone marrow.

BAT is specialized for thermogenesis, a process that generates heat in response to cold or diet-induced factors. It contains a high number of mitochondria and expresses uncoupling protein 1 (UCP1), which uncouples the respiratory chain to produce heat instead of ATP

Location

Found within the bone marrow cavities of long bones and in other skeletal locations. It is distributed throughout the skeleton.

Typically located in discrete depots, such as the interscapular region, neck, and around major blood vessels. BAT has a more limited distribution compared to BMAT.

Cell type

Contains adipocytes, which are cells specialized for fat storage. The differentiation of mesenchymal stem cells in the bone marrow into adipocytes contributes to the formation of BMAT.

Composed of brown adipocytes, which have a unique appearance due to the presence of numerous mitochondria and a high vascularization. These characteristics are essential for BAT's thermogenic function.

Color and apparence

Appears yellowish and resembles white adipose tissue. Its color is due to the accumulation of triglycerides in adipocytes.

Appears brown due to the high concentration of mitochondria and the presence of iron-containing cytochromes. The brown color is a result of increased vascularity and mitochondrial density.

Physiological role

Implicated in bone health, hematopoiesis, and potentially providing a local energy source for bone cells. The exact physiological significance of BMAT is still an active area of research.

Primarily involved in non-shivering thermogenesis, helping to maintain body temperature in response to cold exposure. BAT activity can also impact energy expenditure and metabolic homeostasis.

Association with Health Conditions

Increased BMAT is often associated with conditions such as obesity and certain metabolic disorders. The relationship between BMAT and overall health is complex and continues to be explored.

BAT activation is associated with improved metabolic health and has been investigated as a potential target for treating obesity and related metabolic disorders.

Review 1 comments

Line 155. The sentence is not clear. Is it increasing or decreasing?  

Reply

Dear reviewer thank you for your comment. We explain better the sentence. Below you will find the revise sentence. A strong decrease in brown adipocytes and the corresponding rise in white adipocytes with aging are two of the most noticeable and frequent physiological changes seen in AT

Review 1 comments

What is the effect of cold on BMAT? 

Reply

Dear reviewer thank you for your comment. To better explain this section we have added the following sentence:

“Furthermore, form functionally point of view, data literature reported that the cold ex-posure does not stimulate BMAT glucose uptake in mice or humans, with mouse BMAT also showing no induction of the BAT and beige markers Ucp1, Dio2, Prkaa1 and Metrnl. In addition, some studies suggests that cold exposure may have variable effects on BMAT. Some studies propose that cold exposure could lead to a reduction in BMAT, as the body prioritizes the activation of BAT for thermogenesis. BAT activation is associated with increased energy expenditure, and in some cases, this may result in a decrease in overall adiposity, including BMAT [31-32]. However, the specific impact of cold on BMAT can depend on various factors, including the duration and intensity of cold exposure, individual differences, and the specific mechanisms involved. While there is evidence suggesting a relationship between temperature exposure and adipose tissue dynamics, more research is needed to fully understand the nuances of how cold affects BMAT [33-34].”

Review 1 comments

Page 5: 

Line 185: Does menopause affect BMAT cell number, proliferation, or differentiation? Are there any published studies?

Reply

Dear reviewer thank you for your comment. To better explain this section we have added the following paragraph and relative references:

“Increased expression of pro-inflammatory cytokines, such as TNFα, IL1, IL-6, and receptor activator of RANKL/OPG/RANK system, which controls osteoclast development, is linked to estrogen deficit in the bone microenvironment [35–37]. Reduced endogenous antioxidant production is another effect of estrogen deficiency [38]. When combined with the increased production of reactive oxygen species (ROS) and the senescence microenvironment seen in aging humans and obese mice, these factors can cause a marked increase in oxidative stress in the bone microenvironment [35,39]. Increased bone turnover in response to estrogen deficiency was found in studies involving postmenopausal women and ovariectomized rats. This increased bone resorption and formation was demonstrated by an increase in osteoblast precursors, osteoblast proliferation, and osteoblast number [40,41]. But bone loss results from heightened bone resorption, which is not sufficiently compensated for by increasing levels of bone production. An interesting question that has been raised is whether the biological sex has an impact on the expansion of BMAT during aging? Griffith et al, investigated BMD and BMAT in the lumbar spine of 259 healthy subjects (145 females, 114 males; age range: 62-90 years) using MR spectroscopy of L3 vertebral body, and revealed that in males, BMAT increases gradually throughout life, whereas in females, BMAT increased between 55 and 65 years [42]. BMAT in the vertebral bones in females with the age of more than 60 years increased by nearly 10% higher compared to age-matched males, indicating the positive impact of estrogen deficiency on BMAT expansion. Moreover, research on humans has demonstrated that whereas osteoporosis in men significantly increases vertebral BMAT [43], aging and the loss of steroid hormones make women more susceptible to the detrimental effects of BMAT on trabecular bone loss at the spine and femoral neck, as well as greater loss of spine strength [44]. A review of various clinical and preclinical investigations on the relationship between BMAT and endocrine aging is given.”

Review 1 comments

What is BMD? 

Reply

Dear reviewer thank you for your comment. We explained the acronym BAD in the text.

Review 1 comments

How does sarcopenic Obesity affect BMAT?

Reply

Dear reviewer thank you for your comment. To better explain this section, we have added the following paragraph and relative references:

Sarcopenic obesity refers to a condition characterized by the coexistence of sarcopenia (loss of muscle mass) and obesity (excess body fat). This combination can have various effects on the body, and the interplay between sarcopenic obesity and BAT. Sarcopenic obesity has been associated with changes in adipose tissue distribution, including an increase in visceral adiposity. It's plausible that these changes may extend to the bone marrow, potentially leading to an increase in BMAT content. Sarcopenic obesity is often linked to chronic low-grade inflammation and metabolic disturbances. These factors can influence the bone marrow microenvironment, potentially affecting the composition and function of BMAT. Furthermore, obesity and sarcopenia can both influence hormonal balance. Changes in hormones such as leptin, adiponectin, and inflammatory cytokines may have downstream effects on BMAT and bone metabolism [46].”

Review 1 comments

Page 7: 

Are there any studies linking low vitamin D with a decrease in BMAT differentiation?

Reply

Dear reviewer thank you for your comment. To better explain this section, we have added the following paragraph and relative references:

“An important investigation proposed that the vitamin D receptor (VDR) inhibits adipogenesis through direct inhibition of the expression of PPARγ, the master transcriptional regulator of the adipogenic gene programme [70]. The PPARγ agonist troglitazone reversed the inhibition of adipogenesis by calcitriol, and con- versely, calcitriol antagonized the transactivation capacity of PPARγ in 3T3-L1 cells [70]. VDR overexpression experi- ments showed that even unliganded VDR could inhibit PPARγ transactivation activity, by competing for binding to their common heterodimer partner RXR [70]. Numerous studies have also demonstrated the complementary effects of PPARγ2 and calcitriol in bone marrow and BMSCs, which is consistent with this mechanism. In human bone marrow BMSCs, calcitriol decreased the expression of PPARγ2 [71]. In contrast, the PPARγ antagonist BADGE increased the expression of VDR in bone, enabling calcitriol to enhance the anti-adipogenic effects of BADGE in bone marrow [72]. The expression of PPARγ was upregulated in cultured bone marrow stem cells (BMSCs) derived from VDR null mice. Additionally, there was an increase in adipogenesis and higher expression levels of DKK1 and SFRP2, which are inhibitors of the pro-osteogenic canonical Wnt signalling pathway. In wild-type BMSCs, calcitriol was shown to downregulate the expression of these Wnt inhibitors in both the absence and presence of adipogenic inducers [56]. The selection between osteoblastogenesis and adipogenesis is regulated by the reciprocal inhibition of PPARγ2 and the Wnt pathway [73]. Therefore, vitamin D may restrict BMAT growth through the inhibition of PPARγ2 and the promotion of Wnt signaling.”

Review 1 comments

Please remove number 71 from the text

Reply

We deleted the number 71

Review 1 comments

Does high-fat diet result in high BMAT? Is there any effect of bone resorption on the isolation of BMAT?

Reply

Dear reviewer thank you for your comment. To better explain this section, we have added the following paragraph and relative references:

“A recent study report that HFD-induced obesity in mice leads to increased BMAT formation, enhanced AD differentiation of BM-MSC, and reduced bone mass. Interestingly, BMAT, in contrast to extramedullary adipocytes, did not exhibit a pro-inflammatory phenotype and maintained its insulin responsiveness [74]. This study suggests that BMAT responds to metabolic stimuli and acts as a depot for storing extra energy during HFD when peripheral AT is incapable of mediating this function. Different study observed that HFD increased BMAT formation as evidenced by increased BMAT volume, adipocyte number, and size [75,76]. Other investigations have reported inconsistent results regarding the impact of HFD on bone mass and bone quality. Some studies reported no change or increased or decreased bone mass with lower bone strength [77,78]. These discrepancies can be explained by variations in experimental models, e.g., mouse strain, sex, length/type/composition of diet (60% versus 45% fat or 10% corn oil), and possibly evaluation methods. However, a recent study of Bornstein and colleagues reported HFD induced reduction in the cortical and trabecular parameters in the long bones along with increased BMAT volume, suggesting that bone mass reduction is a biological consequence of coherent mechanisms of cellular and molecular changes relevant for studies of bone fragility in human obesity [74].”

Review 1 comments

Page 8:

Are there any studies showing EVs formed from BMAT? Are they similar to EVs isolated from AT?

Reply

Dear reviewer thank you for your comment. To better explain this section, we have added the following paragraph and relative reference:

“The EVs are small membrane-bound particles released by cells, and they play a crucial role in intercellular communication. These vesicles can carry various bioactive molecules, such as proteins, lipids, and nucleic acids. Studies have investigated extracellular vesicles derived from bone marrow and adipose tissue for their potential roles in communication between cells and tissues. Bone Marrow Adipose Tissue, being a specialized type of adipose tissue located within the bone marrow, is known to influence various physiological processes, including hematopoiesis and skeletal homeostasis [90].”

Reviewer 2 Report

Comments and Suggestions for Authors

In this review Moscatelli and colleagues examined the relation between the MSCs from the BM and obesity. The review starts by defining the multipotential of MSCs and its importance in bone maintenance. Then the authors define the genetic factors regulating the differentiation of stem cells within BM, to continue defining the AT types and their main differences as well as the characteristics of the AT in pathological conditions. Finally, the authors discuss the relation of obesity and AT accumulation in the recruitment and activation of immune cells, or inflammation to finally conclude.  The review is adequately structured, and is comprehensive in the present form. The topic is in time and interesting for the IJMS audience, and the references are updated. The English style may be improved but it its correct and readable, I personally enjoy the reading and it was very informative.  As minor comments I consider that figure 1 can be improved cause is very simple and lacks novelty, perhaps many of the ideas in the text can be summarized in an attractive figure, I’m sure that this may improve the appreciation of the audience.

Figure 1 legend:                may be improved

74           HSC acronym was previously stated (45) same for BMSCs (75)

130        …furthermore, WAT and BAT coexist in a number of 130

Places... Please specify

136        In vitro, in vivo and other Latinisms must be italicized

Author Response

Reviewer 2 comments:

In this review Moscatelli and colleagues examined the relation between the MSCs from the BM and obesity. The review starts by defining the multipotential of MSCs and its importance in bone maintenance. Then the authors define the genetic factors regulating the differentiation of stem cells within BM, to continue defining the AT types and their main differences as well as the characteristics of the AT in pathological conditions. Finally, the authors discuss the relation of obesity and AT accumulation in the recruitment and activation of immune cells, or inflammation to finally conclude.  The review is adequately structured and is comprehensive in the present form. The topic is in time and interesting for the IJMS audience, and the references are updated. The English style may be improved but it its correct and readable, I personally enjoy the reading and it was very informative.  As minor comments I consider that figure 1 can be improved cause is very simple and lacks novelty, perhaps many of the ideas in the text can be summarized in an attractive figure, I’m sure that this may improve the appreciation of the audience. 

Reply

Dear reviewer, thank you for your valuable suggestions which will be fundamental to improving our manuscript.

Reviewer 2 comments:

Figure 1 legend:                may be improved.

Reply

Dear reviewer, thank you for your comment. We have improved figure legend. Following you will find the new caption:

“Figure 1. The figure shows the Multifaceted Connections of Mesenchymal Stem Cells - Unveiling Their Regenerative, Immunomodulatory, and Tissue Engineering Capabilities in Biomedical Research. (MSC: mesenchymal stem cells).”

Reviewer 2 comments:

74           HSC acronym was previously stated (45) same for BMSCs (75)

Reply

Dear reviewer, thank you for your comment. The acronym HSC is specified on line 78 of the new version of the manuscript. The acronym BMSCs is specified on line 123 of the new version of the manuscript

Reviewer 2 comments:

130        …furthermore, WAT and BAT coexist in a number of 130

Places... Please specify

Reply

Dear reviewer, thank you for your comment. We have explained better this paragraph. You will find the version below:

Adipose tissue (AT) is a very heterogeneous organ because to its multi-depot dispersion in addition to its existence as distinct AT types, such as brown AT (BAT), white AT (WAT), and "brite" (brown-in-white). The AT depots have been referred to as "mini-organs" with independent traits and functions due to their extreme variability [27]. The BMAT and BAT are two distinct types of adipose tissues with different functions, locations, and characteristics. In table 1 were reported the key differences between them (Table 1).

Reviewer 2 comments:

136        In vitro, in vivo and other Latinisms must be italicized 

Reply

Dear reviewer, thank you for your comment. Following your suggestion we modified these words.

Round 2

Reviewer 1 Report

Comments and Suggestions for Authors

No comments

Comments on the Quality of English Language

None

Author Response

Dear Reviewer,

Thank you for taking the time to review our manuscript. Your work was fundamental to its improvement. Not having found specific comments, we proceeded to review the grammar of the manuscript and its formatting. Thank you for your work